# *Galactomyces* Ferment Filtrate Potentiates an Anti-Inflammaging System in Keratinocytes

**DOI:** 10.3390/jcm11216338

**Published:** 2022-10-27

**Authors:** Xianghong Yan, Gaku Tsuji, Akiko Hashimoto-Hachiya, Masutaka Furue

**Affiliations:** 1SK-II Science Communications, Kobe Innovation Center, Procter and Gamble Innovation, Kobe 651-0088, Japan; 2Research and Clinical Center for Yusho and Dioxin, Kyushu University Hospital, Fukuoka 812-8582, Japan; 3Department of Dermatology, Graduate School of Medical Sciences, Kyushu University, Fukuoka 812-8582, Japan

**Keywords:** *Galactomyces* ferment filtrate Pitera^TM^, aryl hydrocarbon receptor, NRF2, filaggrin, CDKN2A, caspase-14, claudin, IL-37, IL-33, CXCL14

## Abstract

Skincare products play a crucial role in preventing the dry skin induced by various causes. Certain ingredients can help to improve the efficacy of skincare products. *Galactomyces* ferment filtrate (GFF) is such a functional ingredient. Its use originated from the empirical observation that the hands of sake brewers who deal with yeast fermentation retain a beautiful and youthful appearance. Consequently, skincare products based on GFF are widely used throughout the world. Recent studies have demonstrated that GFF activates an aryl hydrocarbon receptor (AHR) and upregulates the expression of filaggrin, a pivotal endogenous source of natural moisturizing factors, in epidermal keratinocytes. It also activates nuclear factor erythroid-2-related factor 2 (NRF2), the antioxidative master transcription factor, and exhibits potent antioxidative activity against oxidative stress induced by ultraviolet irradiation and proinflammatory cytokines, which also accelerate inflammaging. GFF-mediated NRF2 activation downregulates the expression of CDKN2A, which is known to be overexpressed in senescent keratinocytes. Moreover, GFF enhances epidermal terminal differentiation by upregulating the expression of caspase-14, claudin-1, and claudin-4. It also promotes the synthesis of the antiinflammatory cytokine IL-37 and downregulates the expression of proallergic cytokine IL-33 in keratinocytes. In addition, GFF downregulates the expression of the *CXCL14* and *IL6R* genes, which are involved in inflammaging. These beneficial properties might underpin the potent barrier-protecting and anti-inflammaging effects of GFF-containing skin formulae.

## 1. Introduction

The skin is a vital organ that protects the bodies of terrestrial animals from the effects of dry harsh environments. It also acts as a functional barrier against external mechanical, chemical, and climatological stresses [1,2]. For example, exposure to ultraviolet (UV) rays and environmental pollutants induces varying degrees of oxidative stress in the skin and the subsequent production of proinflammatory cytokines [3,4,5,6]. Low-grade chronic inflammation is a significant risk factor for the type of accelerating aging known as inflammaging [7,8]. An aged skin appearance and a corresponding histological frailty are aggravated in sun-exposed areas of skin compared with those protected from sunlight [9]. Therefore, the inhibition of oxidative stress by daily applications of suitable antioxidants might be beneficial in retarding skin inflammaging induced by various environmental oxidative stress factors [10,11].

The barrier function of skin is mainly provided by its outermost epidermal layer, the stratum corneum or cornified layer [1,2]. The human epidermis is composed of multiple layers of keratinocytes, including basal, spinous, granular, and cornified layers. Keratinocytes proliferate in the basal layer, move up through the spinous and granular layers, and die, but remain functional as corneocytes in the cornified layer, before finally detaching from the skin [1,2]. Corneocytes are the major components of the cornified layer. However, other biological materials, including the extracellular lamellae of lipids, such as ceramides and cholesterol, and various natural moisturizing factors (NMFs), including free amino acids, pyrrolidone carboxylic acids, lactates, glucose, urea, hyaluronic acid, and electrolytes, are essential for maintaining a healthy skin–water balance [1,2,12]. During the differentiation process from the basal to the cornified layer, keratinocytes sequentially produce epidermal differentiation complex proteins, such as involucrin, loricrin, and filaggrin [13]. The integration of these proteins into cytoskeletal keratin fiber is essential for the proper differentiation of keratinocytes into corneocytes [1,2,12,13]. The degradation of filaggrin by proteolytic enzymes, such as caspase-14, in the granular layer is also pivotal in the production of NMFs [1,2,12,13,14,15,16,17].

The differentiation of keratinocyte is coordinately regulated by various transcription factors, including the aryl hydrocarbon receptor (AHR) [18,19], OVO-like 1/2 (OVOL1/2) [20,21,22,23], MYC [22,23], NOTCH1 [22,24], CEBP [25,26], and PPAR [27,28]. The expression or activation of these transcription factors is modulated by certain inflammatory cytokines, phytochemicals, and UV-mediated oxidative stress [29,30,31,32,33,34,35,36,37,38]. For instance, the expression of filaggrin is downregulated by the interleukins IL-4 and IL-13, which are pathogenic for atopic dermatitis, as well as by IL-17A, which is pathogenic for psoriasis [29]. These might contribute, at least in part, to the dry barrier-impaired skin lesions in atopic dermatitis and psoriasis [39].

In general, dry barrier-impaired skin exhibits a decrease in skin hydration and an increased rate of transepidermal water loss (TEWL) [39]. The topical application of a moisturizer increases skin hydration and decreases TEWL [39,40]; therefore, skin moisturization is recommended as a basic treatment, especially for atopic dermatitis and senile xerosis [41,42,43,44]. It is known that skin moisturization is an important factor in facial skin’s ability to maintain a youthful and healthy appearance [45]. Moreover, antioxidative moisturizers can decrease facial redness and reduce pore dilation [46].

## 2. Moisturizers and Their Ingredients 

Moisturizers essentially consist of various functional agents, including occlusive materials and humectants. Occlusive materials, such as petrolatum and lanolin, are hydrophobic and prevent the evaporation of water from the skin by coating its surface with a water-repellent layer that interferes with the bidirectional movement of water across the skin. Petrolatum is a classic example of an occlusive agent that reduces water loss through the epidermis by nearly 99% [47].

Humectants, such as urea, glycerin, and α-hydroxy acids, are compounds that attract and bind water. They can draw water from the deeper epidermis and dermis [48,49]. Moisturizers commonly contain both occlusives and humectants to increase skin hydration and decrease TEWL [50]. In addition to the basic occlusives and humectants, recent advancements in skin biology point to the beneficial potential of topical applications of ceramides or NMFs in upregulating the skin-barrier function [51,52,53].

PITERA^TM^, a specialized *Galactomyces* ferment filtrate (GFF), is a functional ingredient present in multiple skincare formulations that are used worldwide. Historically, research on GFF began from the empirical observation that elderly sake brewers had wrinkled faces, while their hands, which were in constant contact with the sake fermentation process, retained a soft and youthful appearance.

GFF-containing moisturizers are capable of increasing skin hydration and reducing TEWL [46,54]. Clinical trials have also shown that the topical application of GFF Pitera™ improves intraday fluctuations in facial redness, skin roughness, and hair pore size [46]. Mask usage aggravates intraday fluctuations in these facial skin conditions [54]. Mask-induced exacerbation of fluctuations in redness and pore size is also ameliorated by topical treatment with GFF [54]. Although the action mechanisms of GFF are not fully understood, it has been demonstrated to operate as a potent antioxidative AHR agonist [10,55,56]. 

## 3. Activation of AHR-Filaggrin Axis by GFF

AHR is a ligand-dependent transcription factor that is pivotal in upregulating the expression of filaggrin and other differentiation complex proteins in the epidermis [18,19]. In its steady, nonstimulated condition, AHR resides in the cytoplasm of keratinocytes [55]. Upon stimulation by GFF, activated AHR translocates into the nucleus from the cytoplasm (Figure 1), where it upregulates the expression of filaggrin [55]. 

GFF-mediated filaggrin upregulation is AHR-dependent because it is abrogated by the knockdown of AHR [55] (Figure 2). 

The interleukins IL-4 and IL-13 reduce the expression of filaggrin [29,57] and weaken the permeability barrier of keratinocytes [58]; these effects might be responsible for the pathogenic actions of IL-4 and IL-13 in atopic dermatitis [29,59]. Notably, GFF is capable of counteracting the IL-4- and IL-13-mediated downregulation of filaggrin expression [55]. In addition to the upregulation of filaggrin, GFF upregulates the expression of loricrin and counteracts the IL-4-induced inhibition of loricrin expression [55]. A similar AHR-mediated action has also been confirmed to occur with various phytochemicals, such as extracts of *Houttuynia cordata* [60], *Opuntia ficus-indica* [61], and *Artemisia princeps* [62], which are widely used as folk medicines or as cosmetic ingredients. 

The traditional dermatological remedies coal tar and soybean tar glyteer are active AHR ligands and upregulate the expression of filaggrin [63,64]. Both coal tar and glyteer are known to be effective in the treatment of atopic dermatitis, psoriasis, and other inflammatory skin diseases [63,64]. Tapinarof is a recently discovered natural AHR agonist that stimulates the AHR–filaggrin axis [65,66]. Recent clinical trials revealed that the topical application of tapinarof efficiently reduces the skin symptoms of atopic dermatitis and psoriasis [67,68]. Therefore, certain AHR agonists are called therapeutic AHR-modulating agents (TAMAs), and are recognized as promising treatments for inflammatory skin diseases [69]. Considering its active AHR-stimulating potency, GFF is a functional moisturizing ingredient with TAMA-like activity.

## 4. Antioxidative Properties of GFF

The skin is continuously exposed to various oxidative stressors, such as UV radiation, environmental pollutants, and inflammatory cytokines such as tumor necrosis factor-α (TNF-α) [64,70,71,72,73]. These oxidative stressors generate reactive oxygen species (ROS) in skin cells. To ameliorate oxidative damage, excess amounts of ROS require neutralization by an antioxidative system. Nuclear factor erythroid-2-related factor 2 (NRF2) is the antioxidative master transcription factor [10]. Like AHR, nonstimulated NRF2 is mainly located in the cytoplasm of keratinocytes [72,74]. Upon stimulation, the activated NRF2 translocates from the cytoplasm into the nucleus, where it upregulates the transcription of genes for antioxidative enzymes such as glutathione peroxidase 2 (GPX2), NAD(P)H quinone oxidoreductase 1 (NQO1), and heme oxidase 1 (HMOX1), which are responsible for neutralizing excess ROS [56,72,75,76]. For example, GPX2 is known to play a critical role in preventing UVB-mediated carcinogenesis in keratinocyte [75]. 

In addition to its AHR-stimulating properties, GFF induces nuclear translocation of NRF2 in the cytoplasm (Figure 3) and upregulates the expression of GPX2, NQO1, and HMOX1 [10,56,74,76,77]. 

GFF significantly ameliorates both TNF-α-induced [10] and UVB-induced [76] production of ROS in human keratinocytes. IL-13 is also an ROS-inducing cytokine [78]. IL-13-induced ROS production in keratinocytes is also inhibited by GFF (Figure 4). 

Secretory leukocyte peptidase inhibitor (SLPI) is another epidermal differentiation and desquamation maker that is upregulated by NRF2 activation [79]. The increased expression of SLPI is known to impede infection by human papilloma virus by blocking its entry into keratinocytes [80]. Notably, it is reported that GFF upregulated the expression of the *SLPI* gene [81].

## 5. Downregulation of Senescence by GFF

Oxidative stress is also a factor in senescence, as it induces the production of proinflammatory cytokines [6,71], which are involved in inflammaging [7,8,82]. Therefore, antioxidants might have anti-inflammaging action. Senescent keratinocytes have a 30-fold-higher intracellular ROS concentration compared with those in the nonsenescent growth phase [83]. It is known that the senescent cells accumulate an intracellular cyclin-dependent kinase inhibitor 2A (CDKN2A or p16INK4A), which induces cell-cycle arrest [83,84,85]. The epidermal atrophy seen in the elderly is associated with an increase in CDKN2A+ senescent keratinocytes [84]. Transcriptomic profiling of biopsied human skin from the face, arm, or buttock in various age groups has demonstrated that the expression of *CDKN2A* increases with age, especially in skin samples from sun-exposed facial and arm epidermis [9]. 

GFF upregulates the expression of the antioxidative enzyme GPX2 in keratinocytes, which is abrogated by NRF2 knockdown [56]. In contrast, GFF significantly decreases CDKN2A expression in keratinocytes [56]. Notably, the GFF-induced downregulation of CDKN2A is also NRF2-dependent. Therefore, daily applications of a GFF-containing skincare product might be beneficial in preventing the aging process by downregulating CDKN2A expression by upregulating GFF-NRF2-GPX2 axis. GFF-induced NRF2 activation is also observed in melanocytes [77] and, probably, in macrophages [86]. The expression of the antioxidative enzymes HMOX1 and NQO1 is upregulated in GFF-treated melanocytes in an NRF2-mediated fashion [77]. In parallel, UVB-induced ROS generation in melanocytes is alleviated in the presence of GFF [77].

In certain AHR agonists, the activation of NRF2 is mediated, at least in part, through AHR activation [64,87,88]. GFF is also known to activate the AHR–NRF2 signaling pathway, because knockdown of AHR partially ameliorates the GFF-NRF2-mediated induction of NQO1 [76]. 

In addition, it should be noted that the aforementioned beneficial AHR agonists, tapinarof [65], coal tar [63], glyteer [64], *Houttuynia cordata* extract [60], *Opuntia ficus-indica* extract [61], and *Artemisia princeps* extract [62] are all potent inducers of antioxidative enzymes through NRF2 activation. These facts suggest that dual agonists for AHR and NRF2 might be suitable as cosmetic ingredients to maintain the epidermal barrier and to protect against oxidation-induced inflammaging.

## 6. Enhanced Expression of Caspase-14 by GFF

Caspase-14 is preferentially expressed in the granular and cornified layer of skin. It is a cysteinyl aspartate-specific protease that is involved in the degradation of filaggrin into NMFs [89]. *Caspase-14*-deficient mice exhibit a substantial reduction in amounts of NMFs such as urocanic acid and pyrrolidone carboxylic acid [17]. The skin of *Caspase-14*-deficient mice is rough, with decreased skin hydration and increased TEWL, underscoring the important role of caspase-14/filaggrin in maintaining skin moisturization [90]. In addition, *Caspase-14*-deficient mice are more susceptible to UVB-induced phototoxicity [90]. Moreover, IL-4 is known to inhibit the synthesis of caspase-14, which might contribute to barrier disruption in atopic dermatitis [89].

Notably, the expression of caspase-14 is strongly upregulated in GFF-treated keratinocytes [91] (Figure 2). An enhancing activity for caspase-14 expression has been reported for other cosmetic ingredients and natural phytochemicals [92,93,94]. The physiocosmetic implications of caspase-14 upregulation in maintaining skin homeostasis remain uncertain; however, targeting caspase-14 might be a promising strategy for discovering new beneficial ingredients for healthy skin [95,96]. 

## 7. Upregulation of Tight Junction Molecules by GFF

Tight junctions are cell–cell junctions that connect neighboring cells closely [97]. In the human epidermis, mature tight junctions are located in the granular layer and play a crucial role in the paracellular permeability barrier to water, solutes, and high-molecular-weight materials [97]. The tight junction is a protein complex composed of the claudin family, occludin, and other plaque proteins, such as ZO-1 [97,98]. Among the tight-junction molecules, claudin-1 and claudin-4 are critical for maintaining the permeability barrier; this is usually evaluated by measurements of the transepithelial electric resistance (TER) for the transport of ions and of labeled-tracer permeability for the transport of high-molecular-weight substances [98,99,100]. A high TER and a low labeled-tracer permeability indicate a stronger zipper function of the tight junction [98,99,100,101]. Mice with a complete *Cldn1* deficiency die in the first day after birth due to an increased TEWL and a leaky skin [99]. Human subjects lacking *CLDN1* suffer from neonatal ichthyosis-sclerosing cholangitis syndrome, a very rare ichthyosis with severe permeability impairment [102]. Like that of claudin-1, the downregulation of claudin-4 in keratinocytes treated with ochratoxin A is known to promote dysfunction of the epidermal permeability barrier [98].

Notably, GFF increases the expression of claudin-1, claudin-4, occludin, and ZO-1 in keratinocytes [55,81,100] (Figure 2). In parallel, GFF also enhances cell–cell attachment in cultured keratinocytes and augments the permeability barrier, as assessed by an increase in TER [100]. The knockdown of AHR partially downregulates the expression of occludin, whereas it does not affect the expression of claudin-1 or claudin-4 [55]. These results indicate that the GFF-mediated upregulation of claudin-1 and claudin-4 is independent of AHR activation. The GFF-induced upregulation of claudin-1 and claudin-4 might contribute to the moisturizing effects of GFF-containing skin formulae [46,54].

## 8. Increased Production of Antiinflammatory Cytokine IL-37 by GFF

Epidermal keratinocytes are a rich source of the IL-1 family of cytokines, including IL-33 and IL-37 [103,104,105]. IL-33 is a proinflammatory or proallergic cytokine that is overexpressed in keratinocytes derived from a tape-stripped barrier-disrupted epidermis [106]. IL-33 stimulates antigen-presenting cells and shifts naïve T cell differentiation toward type 2 T helper cells, which produce proallergic IL-4 and IL-13 [107,108]. House-dust-mite allergen, a major allergen associated with atopic diseases, activates keratinocytes through toll-like receptor 6 and induces IL-33 production [109]. In addition, house-dust-mite allergen has a protease activity and cleaves the keratinocyte-derived IL-33 to a mature active form [110].

In contrast to IL-33, IL-37 is an antiinflammatory cytokine [111]. IL-37 can inhibit the proinflammatory process induced by a wide range of stimuli, including toll-like receptor, IL-1, and IL-33 [112,113]. The human *IL-37* homologue is found in many mammals, though not in the mouse or chimpanzee [111,114,115].

Notably, GFF upregulated IL-37 expression in human keratinocytes in an AHR-dependent fashion [116] (Figure 2). The GFF-induced IL-37 is biologically active, because it downregulates the expression of IL-33 in keratinocytes [116]. Moreover, the GFF-AHR-IL-37-induced downregulation of IL-33 is canceled by the knockdown of either AHR or IL-37 [116]. Like GFF, the therapeutic AHR agonist tapinarof decreases IL-33 production in keratinocytes through the AHR-IL-37 axis [116]. These results agree with our previous finding that the IL-33 expression is downregulated through activation of the AHR-OVOL1 axis [117] and further point to a critical role of AHR in inducing IL-37 expression in human keratinocytes. GFF-mediated IL-33 downregulation might also be attributable to GFF-mediated claudin-1 upregulation, because IL-33 inhibits the expression of claudin-1 [118].

Notably, GFF is likely to potentiate other antiinflammatory systems. GFF was proved to strongly inhibit the expression of the *CXCL14* gene for the chemokine (C-X-C motif) ligand 14 in keratinocytes [81]. CXCL14 is a potent chemoattractant of immune cells, especially monocytes and dendritic cells [119,120]. It is also known to be closely related to inflammaging [121,122]. GFF also downregulated the expression of the *IL6R* gene for the interleukin-6 receptor [81]. The proinflammatory cytokine IL-6 is related to eczematous dermatitis [123] and is linked to inflammaging [7,8]. Thus, GFF-mediated downregulation of *CXCL14* and *IL6R* might contribute to a retardation of the progress of inflammaging in keratinocytes.

## 9. Conclusions

Recent advances in skin biology not only provide rational evidence and a strategy for understanding the beneficial effects of cosmetic ingredients, but can also help to improve the efficacy of skincare products in treating various skin complaints. Ceramide- and NMFs-containing skincare products are good examples of treatments that significantly alleviate signs of dry skin, even in inflammatory skin diseases [51,52,53]. GFF is a functional cosmetic ingredient whose adoption was inspired by the empirical observation that sake brewers often had youthful and soft skin on their hands. GFF has since been shown to be a potent dual agonist for AHR and NRF2. The GFF–AHR axis upregulates filaggrin production and induces the expression of antiinflammatory IL-37, with subsequent downregulation of the expression of proallergic IL-33. The GFF–NRF2 axis provides a potent antioxidative effect and is probably beneficial in preventing inflammaging, partly through the downregulation of CDKN2A. GFF might also ameliorate inflammaging by downregulating the expression of *CXCL14* and *IL6R*. In addition, GFF increases the expression of caspase-14 and claudins, which are essentially involved in epidermal terminal differentiation and tight-junction maturation.

Medicinal TAMAs, such as tapinarof, coal tar, and glyteer, are potent dual agonists of AHR and NRF2 [63,64,65,69]. Many recent and ongoing clinical studies have demonstrated the therapeutic usefulness of TAMA in atopic dermatitis and psoriasis [67,68,124]. As the biological and functional properties of GFF are similar to those of TAMA, GFF might be categorized as an antioxidative cosmetic AHR-modulating agent. However, the in-depth cellular mechanisms of GFF activity have not been fully elucidated in keratinocytes. Although GFF affects melanocytes [77] and sebocytes [46], the biological significance of GFF with respect to these cell types also remains unclear. Considering its antioxidative, barrier-protecting, and anti-inflammaging/antisenescence properties, GFF can be considered a potent cosmetic agent for preventing and repairing skin damage caused by various external and internal insults, and for maintaining healthy, youthful-appearing skin by retarding skin aging. 

## Figures and Tables

**Figure 1 jcm-11-06338-f001:**
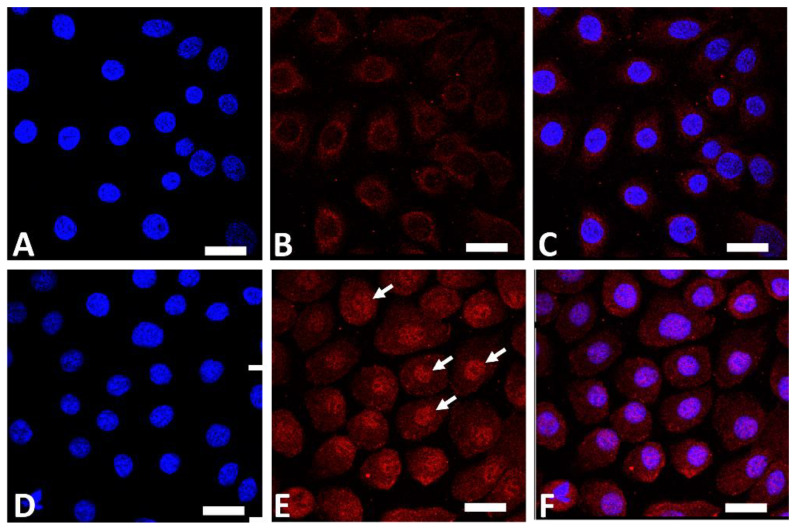
Immunofluorescence staining of AHR (red fluorescence) in human keratinocytes. Nuclei were stained with 4′,6-diamidino-2-phenylindole (DAPI). Nonstimulated control keratinocytes ((**A**); DAPI staining, (**B**); AHR staining, (**C**); merged). GFF-treated keratinocytes ((**D**); DAPI staining, (**E**); AHR staining, (**F**); merged). AHR resides mainly in the cytoplasm in nonstimulated keratinocytes (**B**). GFF induces nuclear translocation of AHR ((**E**), arrows). AHR: aryl hydrocarbon receptor. GFF: *Galactomyces* ferment filtrate. Bar; 25 μm.

**Figure 2 jcm-11-06338-f002:**
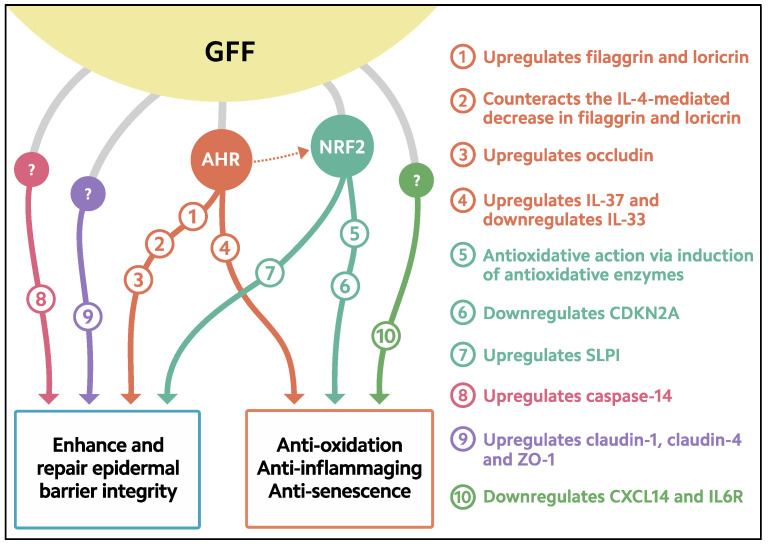
Biological response induced in keratinocytes treated with GFF. GFF: *Galactomyces* ferment filtrate. AHR: Aryl hydrocarbon receptor. NRF2: Nuclear factor erythroid 2-related factor 2. CDKN2A: cyclin-dependent kinase inhibitor 2A. CXCL14: chemokine (C-X-C motif) ligand 14. IL6R: IL-6 receptor. SLPI: secretory leukocyte peptidase inhibitor.

**Figure 3 jcm-11-06338-f003:**
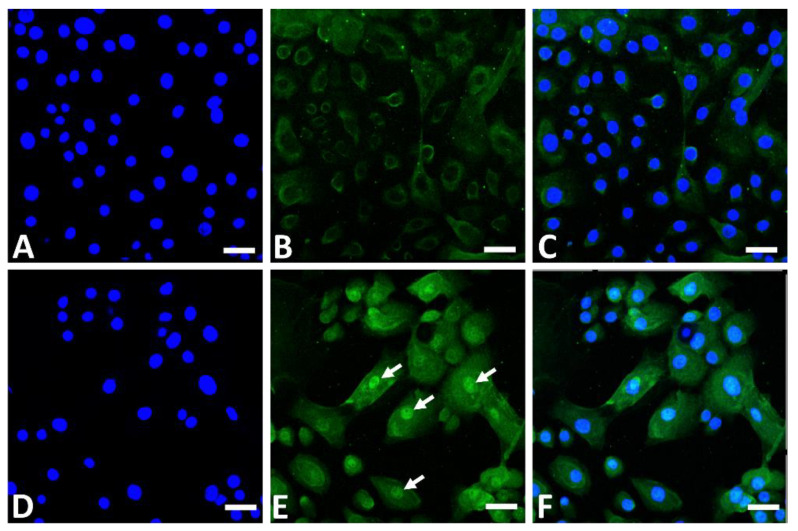
Immunofluorescence staining of NRF2 (green fluorescence) in human keratinocytes. Nonstimulated control keratinocytes ((**A**); DAPI staining, (**B**); NRF2 staining, (**C**); merged). GFF-treated keratinocytes ((**D**); DAPI staining, (**E**); NRF2 staining, (**F**); merged). NRF2 mainly resides in the cytoplasm in nonstimulated keratinocytes (**B**). GFF induces nuclear translocation of NRF2 ((**E**), arrows). DAPI: 4′,6-diamidino-2-phenylindole. NRF2: nuclear factor erythroid-2-related factor 2. GFF: *Galactomyces* ferment filtrate. Bar; 25 μm.

**Figure 4 jcm-11-06338-f004:**
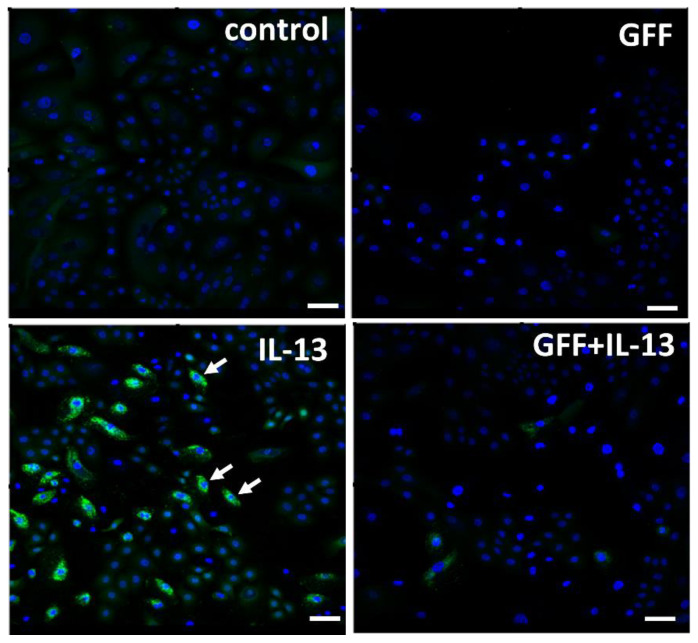
Immunofluorescence staining of ROS (green fluorescence) in human keratinocytes. Nuclei were stained with DAPI. ROS production is minimized in nonstimulated control keratinocytes. IL-13 induces significant ROS production (arrow), which is ameliorated in the presence of GFF. Bar; 25 μm. DAPI: 4′,6-diamidino-2-phenylindole. ROS: reactive oxygen species. GFF: *Galactomyces* ferment filtrate.

## Data Availability

Not applicable.

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
