# Peer review of "Galactomyces Ferment Filtrate Potentiates an Anti-Inflammaging System in Keratinocytes"

_jcm, 2022, doi:10.3390/jcm11216338_

Round 1

Reviewer 1 Report

The authors have performed a comprehensive review of the antiinflammaging effects of Galactomyces ferment filtrate in keratinocytes. The manuscript is well written and informative.

Minor concern:

Suggest use a concise title to make the topic clearer.

Author Response

Reply to the Reviewer 1

The authors have performed a comprehensive review of the antiinflammaging effects of Galactomyces ferment filtrate in keratinocytes. The manuscript is well written and informative.

→ Thank you very much for your encouraging evaluation.

Minor concern:

Suggest use a concise title to make the topic clearer.

→ Thank you very much for your helpful comment. According to your comment, we amended the title as follows; “Galactomyces ferment filtrate potentiates an antiinflammaging system in keratinocytes”

Thank you very much again for your valuable review. We hope the revised article is now suitable for publication in JCM.

Reviewer 2 Report

This is a review article which describes the role of a cosmetic ingredient as antiinflamatory. Understanding the response of keratinocytes to moisture is key in therapeutics to prevent oxidation, inflammation and ageing of the skin. The authors have clearly described the activation routes involved in this process, but also the antioxidative properties. Overall, the information reported in this review is useful. However, the only thing I'm missing is where the figures have been obtained. Did the authors take those images? Were they taken from the literature? If so, please reference accordingly

Author Response

Reply to the Reviewer 2

This is a review article which describes the role of a cosmetic ingredient as antiinflamatory. Understanding the response of keratinocytes to moisture is key in therapeutics to prevent oxidation, inflammation and ageing of the skin. The authors have clearly described the activation routes involved in this process, but also the antioxidative properties. Overall, the information reported in this review is useful. However, the only thing I'm missing is where the figures have been obtained. Did the authors take those images? Were they taken from the literature? If so, please reference accordingly.

→ Thank you very much for your encouraging comments. The figures are our original pictures which we re-obtained by our experiments in order to facilitate or smoothen the readers’ understandings. Of course, the content of all figures were already published using different figures and the references were suitably cited in appropriate sentences in the text. For example, the similar results as Figure 1 were published in reference 55 (line 120-123).

Thank you very much again for your valuable review. We hope the revised article is now suitable for publication in JCM.

To the Reviewer 3

The work entitled “Galactomyces ferment filtrate, an antioxidative AHR-modulating cosmetic ingredient, might potentiate an antiinflammaging system in keratinocytes” (jcm-1946202) has been designed and conducted well, with a remarkable conclusion.

However, the authors should address some issues before considering the manuscript for publication (minor revision).

The manuscript should be revised for linguistic, grammatical, word spacing, and style errors. The format of the whole manuscript and the reference style must comply with the guidelines of the Journal.

Reviewer 3 Report

The work entitled “Galactomyces ferment filtrate, an antioxidative AHR-modulating cosmetic ingredient, might potentiate an antiinflammaging system in keratinocytes” (jcm-1946202) has been designed and conducted well, with a remarkable conclusion.

However, the authors should address some issues before considering the manuscript for publication (minor revision).

The manuscript should be revised for linguistic, grammatical, word spacing, and style errors. The format of the whole manuscript and the reference style must comply with the guidelines of the Journal.

Author Response

Reply to the Reviewer 3

The work entitled “Galactomyces ferment filtrate, an antioxidative AHR-modulating cosmetic ingredient, might potentiate an antiinflammaging system in keratinocytes” (jcm-1946202) has been designed and conducted well, with a remarkable conclusion.

However, the authors should address some issues before considering the manuscript for publication (minor revision).

→ Thank you very much for your encouraging comments.

The manuscript should be revised for linguistic, grammatical, word spacing, and style errors. The format of the whole manuscript and the reference style must comply with the guidelines of the Journal.

→ Thank you very much for your helpful comments. According to your comments, the article was edited by native English editor as certified below. We also re-checked the reference style to fit into the JCM format.

Thank you very much again for your valuable review. We hope the revised article is now suitable for publication in JCM.
